# Tooth Complications after Orthodontic Miniscrews Insertion

**DOI:** 10.3390/ijerph20021562

**Published:** 2023-01-14

**Authors:** Angelo Michele Inchingolo, Giuseppina Malcangi, Stefania Costa, Maria Celeste Fatone, Pasquale Avantario, Merigrazia Campanelli, Fabio Piras, Assunta Patano, Irene Ferrara, Chiara Di Pede, Anna Netti, Elisabetta de Ruvo, Giulia Palmieri, Vito Settanni, Vincenzo Carpentiere, Gianluca Martino Tartaglia, Ioana Roxana Bordea, Felice Lorusso, Salvatore Sauro, Daniela Di Venere, Francesco Inchingolo, Alessio Danilo Inchingolo, Gianna Dipalma

**Affiliations:** 1Department of Interdisciplinary Medicine, University of Bari “Aldo Moro”, 70124 Bari, Italy; 2PTA Trani-ASL BT, Viale Padre pio, 76125 Trani, Italy; 3UOC Maxillo-Facial Surgery and Dentistry, Department of Biomedical, Surgical and Dental Sciences, School of Dentistry, Fondazione IRCCS Ca’ Granda, Ospedale Maggiore Policlinico, University of Milan, 20100 Milan, Italy; 4Department of Oral Rehabilitation, Faculty of Dentistry, Iuliu Hațieganu University of Medicine and Pharmacy, 400012 Cluj-Napoca, Romania; 5Department of Innovative Technologies in Medicine and Dentistry, University of Chieti-Pescara, 66100 Chieti, Italy; 6Dental Biomaterials and Minimally Invasive Dentistry, Department of Dentistry, University CEU Cardenal Herrera, CEU Universities, C/Santiago Ramón y Cajal, s/n., Alfara del Patriarca, 46115 Valencia, Spain

**Keywords:** tooth complications, insertion, miniscrews, nerve, orthodontic, root

## Abstract

Orthodontic miniscrews (OM) are widely used in modern orthodontic clinical practice to improve skeletal anchorage and have a high safety profile. A complication at the time of OM insertion is tooth root perforation or periodontal ligament trauma. Rarely, OM injury can cause permanent damage, such as ankylosis, osteosclerosis, and loss of tooth vitality. The aim of this work was to analyze potential risks and dental complications associated with the use of OMs. A search of the PubMed, Cochrane, Web of Science, and Scopus databases was conducted without a time limit using the keywords “orthodontic mini-screw” and “dental damage”, resulting in 99 studies. After screening and eligibility, including articles obtained through a citation search, 13 articles were selected. Four studies revealed accidental injuries caused by OM. Most of the damage was localized at the root level and resolved spontaneously with restorative cement formation after prompt removal of the OM, while the pain disappeared. In some cases, irreversible nerve damage, extensive lesions to the dentin–pulp complex, and refractory periapical periodontitis occurred, requiring endodontic and/or surgical treatment. The choice of insertion site was the most important element to be evaluated during the application of OMs.

## 1. Introduction

The use of orthodontic miniscrews (OMs) is a valid ally in the clinical practice of orthodontics, representing a valid support in the control of anchorage during orthodontic treatment, both with fixed and removable appliances [1,2] (Figure 1).

Furthermore, OMs expand treatment options for adolescent and adult patients and are a valuable aid in reducing unwanted forces or increasing the effectiveness of orthopedic appliances, such as hybrid or bone braces [3].

Positioning is the major challenge in applying OMs. The rate of success of this technique is strictly conditioned by the site of insertion, angulations, depth of insertion, bone thickness, and design of the screw, e.g., diameter, dimension, and length [4,5,6,7,8,9,10,11,12].

A recent study evaluated the influence of the OM design of the implant surface on retention and, thus, on the success of the OM. In particular, the authors analyzed pitch, thread depth, and shape, concluding that only the first of these three factors is decisive. Implants with a denser thread show greater stability, while thread depth and thread shape are irrelevant [13].

Surgical customized 3D templates are a valuable support in enhancing the insertion accuracy and stability of OMs, reducing the risk of side-effects and failure [14]. The combined use of Cone Beam CT (CBCT) and the intraoral scan allows digital OMs positioning through a Computer-Aided Design-Computer-Aided Manufacturing (Cad-Cam) dime to guide the insertion during the clinical stage.

Although the use of Temporary Anchorage Devices (TADs) in the digital workflow reduces the occurrence of dental damage, it is still a common occurrence. The use of OMs is associated with several complications, which can occur during their insertion, use, or removal [15,16,17]. In the lower arch, the insertion of OMs without a correct evaluation of anatomic structures might determine serious damage to the mental nerve and the lingual nerve.

No significant data are reported about complications due to the perforation of the maxillary sinus by OMs insertion [18]. On the other hand, tooth root injuries are reported as the most frequent complications of OMs [19,20]. The failure of OMs stability is closely related to its proximity to the tooth root. A radiographic overlap of the body of the OM on the roots of dental elements significantly decreases the success rate from 80% in the maxilla and 75% in the mandible to 35% [21,22,23,24,25,26,27,28].

Based on the distance of the OM from the hard plate of the tooth root, three categories can be recognized: (a) category I, in which the OMs are distant and separated from the tooth root; (b) category II, in which the apex of the screw grazes the hard lamina; (c) category III, in which the body of the screw invades the lamina dura. Categories I and II are characterized by a high success rate (75%), while in category III, the stability of OMs is significantly reduced, especially at the mandibular level (35%). In addition, incorrect positioning of OMs in relation to the tooth root leads to a double complication, namely mobility and failure of the OMs and injury to the tooth element [29].

There are existing reviews about the risks associated with the insertion of OMs related to other structures such as maxillary sinus or palatine artery, while the literature is expected about the complications on teeth. It is necessary for orthodontics to widely understand the potential dental risks that can occur using OMs, to avoid them, and to improve clinical practice [30,31,32]. The aim of this review was to evaluate tooth injuries after OMs insertion, in order to understand potential risks and dental complications associated with the use of OMs [33].

## 2. Materials and Methods

### 2.1. Protocol and Registration

This review was carried out according to the recommendations of the literature search Preferred Reporting Items for Systematic Reviews and Meta-analysis (PRISMA) and recorded in the International Prospective Register of Systematic Review (PROSPERO) with the number CRD42022380972 [34].

### 2.2. Search Processing

We searched without a time limitation in PubMed, Cochrane, Web of Science, and Scopus to discover publications that matched our topic. The following Boolean keywords were included in the search strategy because they closely fit the goal of our examination, which primarily focuses on the damaging effects of OMs on teeth: (“orthodontic mini-screw” AND “tooth damage”) (Table 1).

### 2.3. Inclusion and Exclusion Criteria

The inclusion criteria were (1) studies only on humans; (2) English language, (3) open-access studies; (4) clinical trials or case reports. Studies regarding tooth damage in the presence of OMs, and studies dealing with surgical procedures, complications unconnected to teeth, OMs, or any other aspect of orthodontic care were excluded.

### 2.4. Data Processing

Disagreements between authors on article selection were discussed and resolved.

## 3. Results

The electronic database search identified a total of 99 studies (PubMed n = 45, Cochrane n = 1, Web of Science n = 18, and Scopus n = 35). After removing duplicates, 53 studies remained. Six more pertinent articles were added by looking over the reference list of acceptable publications. A total of 36 articles were excluded because they were off-topic, 9 were not available, and 2 were animal studies. The study selection and process are summarized in Figure 2. After a full-text review, a total of 13 studies met the inclusion criteria. A summary of the characteristics of each study included is presented in Table 2.

## 4. Discussion

Contact between the tooth root and the screw can cause trauma of the periodontal ligament or to the root, and this can lead to root resorption [48,49,50,51] (Figure 3).

If the root is just superficially injured and there is no pulpal involvement, the injury will heal successfully by the deposition of cellular cementum once the contact has been broken [52]. In addition, a faster and complete repair takes place a few weeks after removing the screws [37]. The repair process can start as early as 7 to 10 days after the release of the force. Several studies have confirmed that 75% of the repair might be completed within 8 weeks [38], and others have demonstrated complete repair of the tooth and periodontium in 12 to 18 weeks after removal of the OM [48]. Others have demonstrated complete healing after almost 20 weeks, depending on the severity of the injury [53].

Kadioglu et al. examined the damage to the first premolars root surfaces in patients whose teeth must be extracted to correct a class II 2 division malocclusion [38]. The OM was placed near the roots of the right and left first and second premolars. The teeth of the control group that came into direct contact with the OM were extracted without the chance of repair; these teeth showed extensive damage at the root level. This study demonstrated that the injury in the root surfaces that have come into contact with the OM can be repaired through the reorganization of collagen fibers with no clinical consequences [38].

Ghanbarzadeh et al. studied the histological response of cementum, root dentin, and pulp, caused by intentional injury of the roots with self-tapping and self-drilling OMs [36]. The teeth were extracted eight weeks after the insertion of OMs, for the orthodontic treatment. Reparative cementum formation was observed in 75.4% of the teeth and there was no significant difference between the group with repair and the group with no repair regarding the formation of the restorative cement [36].

The effects of the contact of the OM with the dental root both at insertion and during orthodontic treatment have been described also by Baik et al. In both cases analyzed by the authors, the OMs were removed immediately, with pain relief, and relocated to other sites, without affecting the movement of the tooth and without loosening the screw. Cement contributed to root repair starting at 8 weeks after immediate screw or orthodontic strength removal. After approximately 12 months, there were no residual symptoms, such as ankylosis, discoloration, root canals, or pain. Orthodontic root movement in contact with the OM failed. By promptly removing the OMs, the problem of screw loosening was also avoided [40].

Lim et al. revealed that inappropriate and repeated insertion of OMs causes persistent periodontal inflammation and root damage, leading to pulp necrosis as well as refractory periapical periodontitis that necessitates surgical intervention. Accurate knowledge of the architecture and physiology of the root, tooth pulp, and periodontal tissue, as well as the benefits and restrictions of this technique, is crucial. In addition, a thorough evaluation of the damage done to both the root and the pulp and, finally, a careful consideration of the best treatment option are required [39].

The use of OMs usually never results in long-term problems. However, root injury can occasionally result in consequences such as ankylosis, osteosclerosis, and loss of tooth vitality. Ankylosis can occur if the damaged region is more than 4 mm or 20% of the root surface [37]. The close contact of dental roots appears to be the primary factor contributing to problems of surrounding teeth during implantation of OMs and following orthodontic loading [54]. Depending on the severity of the injury, a root fracture that involves the pulp tissue may cause the tooth to lose its vitality. Pulp vitality is lost when a screw penetrates the root by more than 50% of the screw’s diameter [41,55]. The periapical and surrounding periodontal tissues are destroyed as a result of the pulp necrosis that follows [37]. When the damage of root–screw contact is limited to the periodontal dental ligament (PDL), the injury is likely to be repaired with no further consequence, but if the cementum is mechanically damaged, and the dentin is exposed, multinucleated cells colonize the denuded surfaces and resorption takes place, supported by continuous stimulation [37,38,53]. Due to the persistence of continuous trauma and inflammation, a breach produced by the OM allowed outside pathogens to gain access to the PDL space and bony destruction around the roots, which ultimately resulted in pulp necrosis [35,56]. Radiological and histological testing can show injuries at the roots [37]. The loss of vitality is diagnosed with not being responsive to pulp sensitivity tests, such as cold tests and electric pulp testers. Apical radiography helps to reveal the clinical situation, at the insertion time and after the screw time removal. The insertion takes place in the attached gingiva until the drilling of the cortex [37,55,57,58]. The resulting symptoms are localized pressure and pain sensation, despite the topical anesthesia. The suspected trauma will be checked radiographically, and the clinician will screw out the fixture, in two or three turns to give relief from symptoms [48]. Once the radiographic confirmation of damage has been obtained, a root perforation can be treated through the access cavity. The treatment of a perforation concerns an intracanal approach. In case of failure of the conventional endodontic treatment, a surgical repair is indicated [35,37,43]. The surgical approach involves the creation of a bone window and consequent sealing of the perforation site with material such as mineral trioxide aggregate (MTA). The prognosis strongly depends on the treatment of bacterial infections at the perforation site [35,43].

The endodontic management of tooth damage after the insertion of OMs was analyzed by McCabe et al. The authors showed a case of accidental OM tip fracture during the insertion and included near the tooth (element 1.6) [41]. More than 1 week later, a sinus tract over 1.6 was noted, but the tooth responded positively to sensitivity testing. The retained screw was surgically removed and the 1.6 tooth seemed to be asymptomatic for 9 months until the end of orthodontic treatment when the fistula reappeared. The patient referred a history of pain that was made worse by cold. To percussion, the tooth was just mildly sensitive. The mobility was normal and periodontal probing was 3 mm interproximal to 1 mm buccally and lingually. Intraoral radiography showed a lesion in proximity to mesiobuccally and palatal roots. The endodontic treatment of MB and MB2 needed an apicectomy below the perforation. A 5-year follow-up revealed a favorable outcome with no signs and symptoms. It is crucial to obtain periapical radiographs of the intended implant location. Intraoral radiographs are not typically performed by most orthodontists. However, a two-dimensional picture cannot be used to determine the precise location of the roots. In addition, this report shows the favorable outcome of a iatrogenic perforation caused by the insertion of OMs [41].

Histological examinations, through specific coloration, e.g., blue of toluidine, allow for understanding the mechanisms of damage and repair lining the root, with details about cementum, PDL, and bone [54]. If the defect on the root surface is repaired, histological examination revealed woven bone at the insertion site and positive staining due to the activity of cementoblasts, which starts after the screw removal at the earliest 12 weeks, and lasts up for 18 or 20 weeks [54]. Through high-powered observation, Ahmed et al. found that the newly formed concrete was strongly oriented irregularly in confront to the normal healing and brownish red hemosiderin in pigmentation in several areas of the repaired cement [35]. The inner layer of Dentin Pulp Complex (DPC), following the “S” shaped dentinal tubules, revealed abnormal deposition of pre-dentine along normal uninvolved root surfaces. The DPC exposed an hematophilic area, indicating a more maturing, old, or new predentine and an eosinophilic area referring to a newly deposit predentine. The histological examination detected an increased level of predentine in the site of the OMs insertion and an irregular orientation in the external surface of dentin, while the internal zone presented numerous globular vacuolar spaces with a completely irregular outline. In the external fascia of the predentine, hemosiderin pigments were also detected. The formation of cementum around the dentin is believed to be due to two causes, i.e., the engulfment of cementum particles during the repair process and the differentiation of the periodontal ligament cells into cementum-producing cells. The presence of irregular vacuoles in the internal surface of neodentin indicates an injury of the pulp due to OMs insertion. The hyperproduction of hemosiderin in the predentine is the reflection of abnormal bleeding at the site of the trauma by OMs insertion, which may be related to the release of metal ions, e.g., vanadium and aluminum, and to altered pH [35]. However, the resorption process will end on its own if there is no additional stimulation, and a phase of healing with tissue similar to cementum will begin [37,59]. These issues result from the placement of OM in the alveolar process between roots, which needs a crucial pre-evaluation [25]. During inter-radicular placement, the clinician may change accidentally the angle of insertion, significantly increasing the risk of damage. Indeed, the available space between roots is more in the apical zone, but without attached gingiva. Thus, the clinician must remain in the “safe zones”: allowing 2 mm from the root, at least 1.0 to 5.0 mm [35]. Specifically, in the upper arch, the OM placement over the greater palatine artery increases the risk of palatal root contact and loss of biomechanical control [38,48,60,61]. In the lower arch, we could see nerve involvement, and the clinical recommendation is to use a short retromolar OM, In this case, many authors have reported measurements with screws no longer than 8 mm, and placed them in the buccal retromolar region below the anterior ramus, for the mandibular site [48,62] (Figure 4).

Hwang et al. described the root injury arising from the placement of OMs in a mandibular incisor and the management of this complication [37]. One of the two inserted OMs, between the central and lateral incisors, inserted to obtain anterior sector intrusion pierced the root of the right lateral incisor. After detecting no vitality of the tooth and considering the radiolucency in the apical area on X-ray, apicectomy was performed, closing the perforation through the application of MTA [37,63]. The degree of root injury was related to the degree of injury. When the screw–root interface was restricted to the area of the periodontal ligament, the injury could be healed without further sequelae [37,64]. If the cementum was mechanically destroyed and dentin was revealed, multinucleated cells would colonize the surfaces and resorption would begin. Without additional stimulation, the reabsorption phenomenon stopped spontaneously, and cement-like tissue occurred within 2 to 3 weeks [37,65]. However, an irreversible reaction could have resulted when the affected zone was large or the injury was deep, so endodontic and surgical treatment was performed, and the perforation of the root was treated with MTA [37,66].

The major limitation of this review concerns the quantity and quality of the articles included in the study. Currently, in the literature, the number of studies concerning adverse effects and complications of OM is low and of a low methodological level: most articles analyze a very limited number of cases and do not include adequate control groups.

This review aimed to investigate the damage caused by OMs by focusing only on teeth damage. This choice was made considering that there are already reviews in the literature dealing with the damage of OMs to other tissues in the oral cavity, such as the maxillary sinus, or the general risks due to the insertion of OMs. With the exception to pay more attention during insertion, no other procedure for optimal OMs placement can be obtained. More consisting randomized controlled trials are needed to provide evidence-based guidelines to clinicians.

## 5. Conclusions

Anchorage control is one of the most challenging problems in orthodontics. It is achieved with OMs, which have been used frequently because they prevent unintended tooth movement and reduce the dependence on patient cooperation, increasing the predictability of achieving treatment goals. During orthodontic treatment, the choice of an appropriate site for OMs placement is more important than the insertion technique. Site selection requires recognition of the anatomy of the tooth roots and pulp, the distance between the tooth roots, and the physiology of the structures surrounding the tooth, such as the periodontal complex. Placement of OMs is a critical procedure, and even if preventive measures are taken, such as an apical radiograph before screw placement, root damage can occur. Screw–tooth contact does arise, but the incidence of clinically relevant injury seems to be low. When screw–root contact injury is limited to the periodontal structures, the lesion is usually repaired without further consequences. In many cases, cement healing was observed after the injury was intentionally caused on the tooth, and the two approaches to placing the OMs, self-tapping and self-drilling, showed no relevant differences in the healing process. Healing was achieved by eliminating continuous contact with the tooth and was achieved by prompt removal of the screws or springs tipping of the screw. In case of extensive damage to the dental root and iatrogenic perforations, endodontic and surgical treatment had to be performed to manage this complication. Despite the lack of clinically significant damage in the patients examined, clinicians must be prudent in the placement of OMs. More accurate planning of OMs placement is required to minimize the risk of root damage.

Finally, the results showed that whether the insertion of OMs is performed accurately, dental complications in most cases are minor and sometimes resolve spontaneously. Therefore, the application of OMs was found to be a safe procedure with few adverse effects. This turns out to be important in orthodontics where miniscrews are a very valuable tool for the resolution of even complex cases.

However, considering the low number of patients analyzed in this review, further studies are necessary in the future to investigate and analyze the topic more comprehensively and to draw up a list of evidence-based clinical recommendations.

## Figures and Tables

**Figure 1 ijerph-20-01562-f001:**
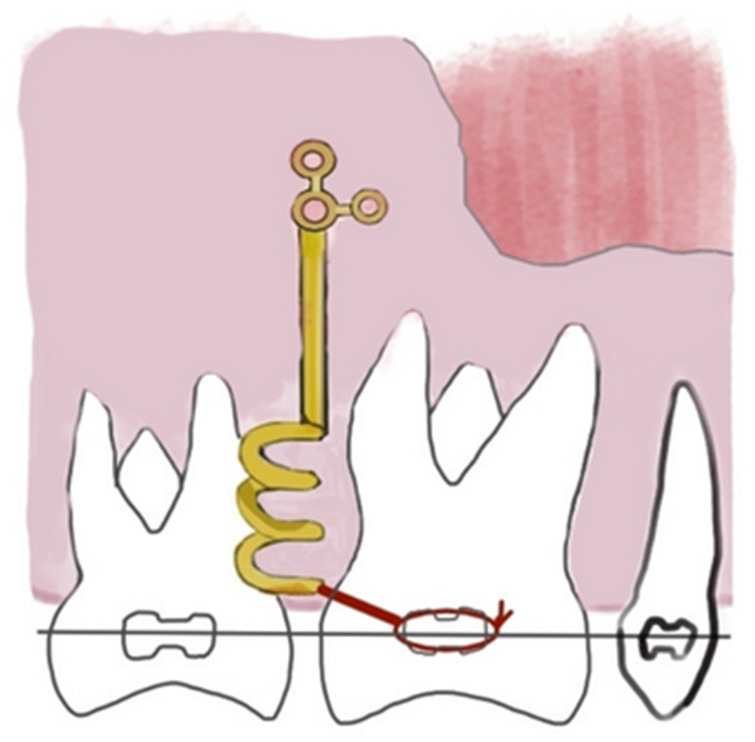
Orthodontic miniscrews (OMs) are anchorage devices in orthodontic treatment.

**Figure 2 ijerph-20-01562-f002:**
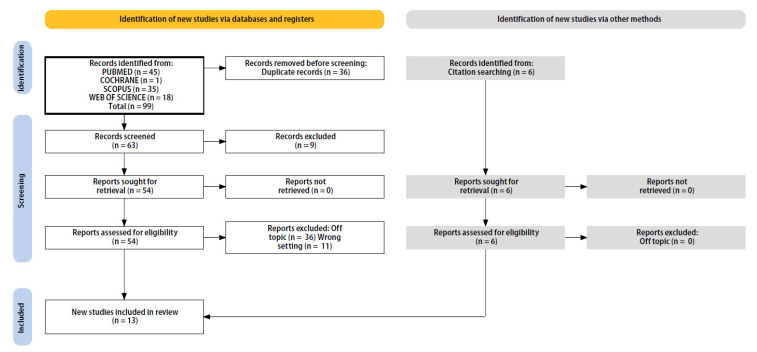
Literature search Preferred Reporting Items for Systematic Reviews and Meta-Analyses (PRISMA) flow diagram and database search indicators.

**Figure 3 ijerph-20-01562-f003:**
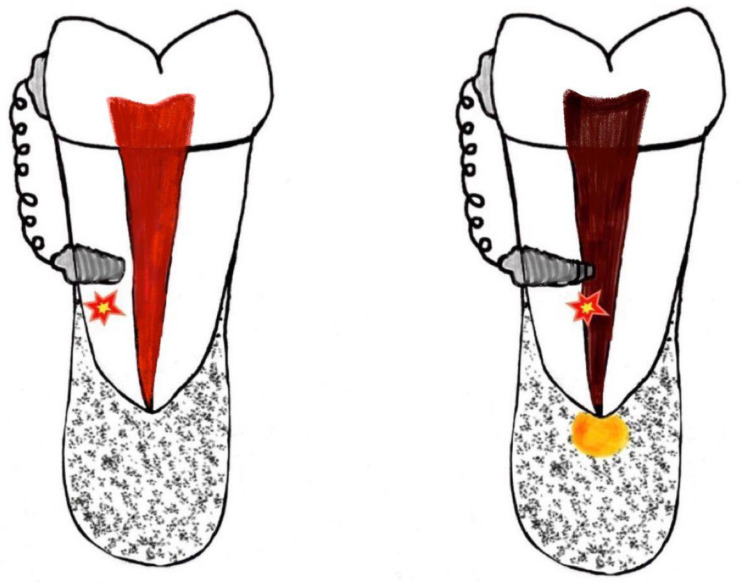
Dental trauma may occur in the insertion of the OMs.

**Figure 4 ijerph-20-01562-f004:**
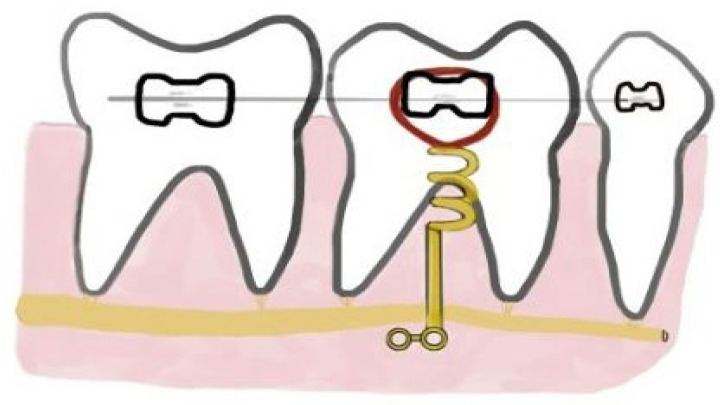
Attention to the insertion of orthodontic miniscrew (OMs) in the presence of the inferior alveolar nerve.

**Table 1 ijerph-20-01562-t001:** Database search indicator.

Articles screening strategy	KEYWORDS: A: “orthodontic miniscrew”; B: “tooth damage”
	Boolean Indicators: (“A” AND “B”)
	Timespan: no limit
	Electronic Database: Pubmed, Cochrane, Web of Science and Scopus

**Table 2 ijerph-20-01562-t002:** Characteristics of the included studies.

Author	Study Design	Sample	Type of Complications	Causes	Healing	Parameters	Recommendations
Ahmed et al., 2016 [35]	Case report	1	Damage of the dentin-pulp complex	Intentional root contact due to OM tip fracture	Tooth extraction after 4 weeks	Spontaneous; Normal pulp vitality test	No difference in insertion technique. Attention to the insertion
Ghanbarzadeh et al., 2017 [36]	Clinical study	14	Response of cementum, root dentin, and pulp with inflammation	Intentional root injury	Tooth extraction after 8 weeks	Spontaneous; Normal pulp vitality test	No difference in insertion technique. Attention to the insertion
Hwang et al., 2011 [37]	Case report	1	Iatrogenic root perforation and chronic apical periodontitis	Accidental injury	No	No spontaneous healing; Negative pulp/cold test	Correct selection of OMs insertion site
Kadioglu et al., 2008 [38]	Clinical study	10	Root resorption	Intentional premolars root contact	Tooth extraction after 4 or 8 weeks	Spontaneous; Normal collagen fiber organization	Attention in insertion
Lim et al., 2013 [39]	Case report	1	Root surface damage	Accidental injury	No	No spontaneous healing	Root canal treatment and surgical intervention; Attention in the insertion; Importance of anatomy
Baik et al., 2014 [40]	Case report	4	Clinical consequences after root contact	Accidental injury	Removed OM immediately after root damage	Spontaneous healing	Attention in insertion
McCabe et al., 2012 [41]	Case report	1	Iatrogenic root perforation	Accidental injury	No	Pulpal necrosis and apicectomy	Attention in insertion
Maino et al., 2007 [42]	Clinical study	2	Root perforation	Intentional injury	The teeth whose roots were forced up against the screws until extraction (group 1 after 30 days of healing and group 2 after 57 days)	Teeth had active resorption lacunae without any indications of healing, according to a histological investigation	Damage to resorptive roots is caused by drill contact and/or TADs. Thereafter, repair the breaking off of contact
Ahmed et al., 2012 [43]	Clinical study	17	Root perforation	Intentional injury	After 12 weeks, reparative cementum forms, resulting in a satisfactory repair	All of the teeth chosen for this study showed good recovery at all stages of the healing process. By the conclusion of week 4, major repairs had been made. By week 4, almost 60% of the teeth showed moderate repair; by week 8, this number dropped to 35%, and by week 12, it was down to 28.6%. By week 12, 70% of all teeth showed good healing	For both periodontal health and miniscrew stability, there should be a minimum of 1 mm between a miniscrew and a root. Therefore, if the gap between the roots is at least 3.5 mm, miniscrews with a diameter of 1.5 mm or less are safe for interradicular insertion
Hourfar et al., 2017 [44]	Retrospective study	284	Loss to pulp sensibility testing of maxillary front teeth after paramedian (3 to 5 mm away from the suture) OMs insertion in the anterior palate.	Age, gender, inclination of upper centrals, dentition status, and insertion position	3 of 284 samples presented loss of vitality	No spontaneous healing; Negative pulp/cold test	The closeness of the implants to the anterior teeth was positively and significantly associated with loss of vitality, even though there was no radiographic evidence of OMI-induced damage to the teeth that lost vitality. The third rugae or the middle plane should be where OMs should be positioned
Yerawedakar et al., 2018 [45]	Clinical study	10	Root perforation	Intentional injury	Extraction at time intervals of one day, three weeks, six weeks, and 12 weeks	Histological investigation showed break in the continuity of cementum after one day. Cementum formation began at early third week, and the thickness gradually increased by the end of 12 weeks. Progressive increase in the thickness of newly formed cementum filled the notch completely at the end of 12 weeks.	Further tooth movement should be prevented for at least a minimum of 12 weeks while the root heals
Shinohara et al., 2013 [46]	Prospective study	50	Root contact	Accidental injury	20% of 147 OMs of the mini-implants in this study contacted adjacent roots	Measuring the mesial and distal distances between the mini-implant and the root allowed researchers to analyze the mesiodistal implantation position. Then, the implant’s inclinations in both the horizontal and vertical	Mini-implants should not make contact with the root of the distal neighboring tooth when being inserted into the right maxillary buccal alveolar bone
Güler et al., 2019 [47]	Clinical study	42	Root contact	Intentional injury	Left premolars were extracted 4, 8, or 12 weeks after OMs root contact. According to micro-CT results, a 12-week healing time might not be long enough to completely repair the roots	Repair process 4, 8, and 12 weeks after TAD contact by micro-TC	In comparison to leaving a TAD in touch with the root in situ without force, force loading from springs on injured roots generated by TAD contact during orthodontic treatment may worsen damage and decrease healing. No more than 12 weeks should pass after forcing reloading on broken roots

## Data Availability

Not applicable.

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
