# Peer review of "Tooth Complications after Orthodontic Miniscrews Insertion"

_ijerph, 2023, doi:10.3390/ijerph20021562_

Round 1

Reviewer 1 Report

This review brings little new to the literature. Please see enclosed pdf for further details

Reviewer 2 Report

We have read with great interest the manuscript with title:” Tooth Complications after Orthodontic Mini-Screws Insertion “.

The manuscript is of interest; however, several criticisms have been found and need to be addressed before resubmission.

1.     A revision by a native English speaker is mandatory before resubmission.

2.     In the abstract: please state clearly the aim of the review, please add the period limit of the search.

3.     I noticed a total of 27 authors, in my opinion, this review can admit 10 authors. This article can have a maximum of 10 authors. Beyond that number, it is really impossible to find a rationale for almost three times as many authors, and in this case, it is a small systematic review with no meta-analysis.

4.     M&M: please add the Prospero registration number.

5.     M&M: please provide the string of the search used on Pubmed, Cochrane, Web of Science and Scopus.

6.     Please provide an updated to 2022 version of the Flowchart of the search. 

7.     4 out of 7 studies are case reports of a single case!!!! suggest to the authors to exclude these studies from the systematic review, in order to preserve a minimum of scientific seriousness. Moreover, in the inclusion criteria it should be added that case reports are included if there are at least 5 clinical cases (!) thank you.

8.     it is requested to add in the results, the total number of patients of the included studies, after deleting the 4 articles reporting 1 clinical case each, the mean age, age range, % of men and women.

9.     Please add these two articles to the discussion section: doi: 10.3390/jcm11185304 and doi.org/10.3390/coatings11121488

10.  Please add a Limitation section at the end of the Discussion.

11.  Conclusions must be brief: this literature review analysed 28 patients. (!!!) Which is a very small number, almost insufficient to draw any kind of valid conclusion. Certainly, it is a documentation that does not allow any evidence-based recommendations to be written. This is the greatest limitation of this manuscript and must be emphasised.

Reviewer 3 Report

This article was based on the review on the Tooth Complications after Orthodontic Mini-Screws Insertion involved 7 selected article with suit the inclusion and exclusion criteria.

Overal the manuscript is well written and had focus on the topic related. Abstarct, introduction parts were satiscify. the method can be improved by adding and decribe how the authors get the aggreements/ disagreements on the selection of the papers/articles.

 The results and discussions and conclusion are ok.suggest to the authors to add the limitation of the study/review and provide the suggestions or recommendations related to the findings.

The references- suggest to follow the guidelines from the journal. the reference cited for no 46 should be no 40 for page 6 line 165 and the rest of the refences cited in the manuscript after that should be corrected accordingly. Please revise the references in the manuscript and the reference sections.

Round 2

Reviewer 1 Report

The manuscript has been improved however I stand by my initial comment that this review brings little new to the existing literature.

Reviewer 2 Report

Dear Authors,

Thank you for making the suggested changes to the text of the manuscript, which certainly increased its quality and scientific value.

In my opinion, the manuscript can be sent, in the present form, to the editorial office for all those operations that are necessary to proceed toward publication.

Best regards
